# Prunin Laurate Derived from Natural Substances Shows Antibacterial Activity against the Periodontal Pathogen *Porphyromonas gingivalis*

**DOI:** 10.3390/foods13121917

**Published:** 2024-06-18

**Authors:** Erika Wada, Chiharu Ito, Mai Shinohara, Satoshi Handa, Miki Maetani, Mayo Yasugi, Masami Miyake, Tatsuji Sakamoto, Ayaka Yazawa, Shigeki Kamitani

**Affiliations:** 1Nutrition Support Course, Graduate School of Comprehensive Rehabilitation, Osaka Prefecture University, Habikino 583-8555, Osaka, Japanayazawa@omu.ac.jp (A.Y.); 2Division of Clinical Nutrition, School of Comprehensive Rehabilitation, Osaka Prefecture University, Habikino 583-8555, Osaka, Japan3428hsm@gmail.com (M.M.); 3Graduate School of Agriculture, Osaka Metropolitan University, Sakai 599-8531, Osaka, Japan; handa-s@toyosugar.co.jp (S.H.); sakamoto@omu.ac.jp (T.S.); 4Graduate School of Veterinary Science, Osaka Metropolitan University, Izumisano 598-8531, Osaka, Japan; shishimaru@omu.ac.jp (M.Y.); masamimiyake@omu.ac.jp (M.M.); 5Department of Nutrition, Graduate School of Human Life & Ecology, Osaka Metropolitan University, Habikino 583-8555, Osaka, Japan

**Keywords:** periodontal disease, antibacterial activity, *Porphyromonas gingivalis*, flavanone

## Abstract

Periodontal disease is an inflammatory disease caused by infection with periodontopathogenic bacteria. Oral care is essential to prevent and control periodontal disease, which affects oral and systemic health. However, many oral hygiene products currently on the market were developed as disinfectants, and their intense irritation makes their use difficult for young children and older people. This study investigated the antibacterial effects of prunin laurate (Pru-C12) and its analogs on periodontopathogenic bacteria, *Porphyromonas gingivalis* (*P. gingivalis*). Pru-C12 and its analogs inhibited in vitro bacterial growth at more than 10 μM and biofilm formation at 50 µM. Among its analogs, only Pru-C12 showed no cytotoxicity at 100 µM. Three of the most potent inhibitors also inhibited the formation of biofilms. Furthermore, Pru-C12 inhibited alveolar bone resorption in a mouse experimental periodontitis model by *P. gingivalis* infection. These findings may be helpful in the development of oral hygiene products for the prevention and control of periodontal disease and related disorders.

## 1. Introduction

Periodontitis is an inflammatory disease of the gingiva and periodontal tissues caused by infection with periodontal pathogens [1,2,3]. The long-term presence of periodontal bacteria in the oral cavity causes inflammation of the gingiva and further destruction of periodontal tissue. The prevalence of periodontal disease is about 45% to 50% of adults in the mild form and becomes more than 60% in people over 65 years old. Severe periodontitis is estimated to affect 11.2% of the world’s adult population [4]. A relationship between periodontopathogenic bacteria and various systemic diseases, such as aspiration pneumonia, osteoporosis, and cardiovascular diseases, has been strongly suggested recently [5,6,7,8,9,10,11,12].

Maintaining dental and oral health is not only essential for chewing food but also serves as a foundation for enjoying a prosperous life through meals and conversations. For elderly individuals, who are among the primary sufferers of periodontal disease, chewing function, which includes the teeth, is crucial for ensuring the quality of life. The primary cause of tooth loss is periodontal disease, which mainly depends on the severity of periodontal classification [13,14].

When periodontal bacteria reside in the gingival crevice for an extended period, inflammation occurs in the gums. As the disease progresses, the supporting alveolar bone of the teeth is absorbed, leading to a decrease in the number of teeth through eventual tooth loss [2,3]. Risk factors for periodontal disease include environmental factors such as smoking, nutrition, and alcohol consumption, host factors such as diabetes and aging, and bacterial factors, with the latter being an essential trigger [15]. Periodontal pathogenic bacteria comprise numerous oral bacteria, classified schematically based on their pathogenicity [16]. Among them, *Porphyromonas gingivalis* (*P. gingivalis*) is frequently detected in deep periodontal pockets and is considered the most critical bacterium in periodontal disease [15]. *P. gingivalis* is a keystone pathogen that causes dysbiosis by disrupting the homeostasis of the indigenous bacterial flora with the host through complementation [17]. *P. gingivalis* is a Gram-negative anaerobic assacharolytic bacterium that possesses a variety of virulent factors, such as cysteine proteinases (gingipains), lipopolysaccharide (LPS), hemagglutinins, and adhesin as fimbriae [18].

Prevention is crucial in managing periodontal disease [16]. Preventive measures involve inhibiting the growth of periodontal pathogenic bacteria and the production of pathogenic factors. However, many commercially available oral hygiene products are primarily developed as disinfectants and tend to be highly irritating. Therefore, it is challenging to use such products for elderly individuals, who are among the primary sufferers of periodontal disease, and for young children who need early preventive measures to reduce the risk of developing periodontal disease. It is desirable to use safe and low-irritation substances, especially for the elderly and young children. The substances derived from natural products may potentially exhibit lower irritation compared to chemically synthesized substances. Hence, this study focused on antimicrobial substances derived from natural products such as flavonoids. Flavonoids are secondary metabolites in almost all plants and possess various physiological activities, including antimicrobial, antioxidant, and anti-inflammatory properties [19]. Prunin is a flavanone glycoside obtained by hydrolyzing naringin from the peels of fruits like grapefruits [20]. Prunin laurate, an ester compound of prunin and plant-based fatty acid lauric acid, has been reported to exhibit antimicrobial effects against lactic acid bacteria, spoilage bacteria, especially *Listeria monocytogenes* [20], and various Gram-negative bacteria [21] such as *Escherichia coli*. However, the impact of prunin on *P. gingivalis*, a Gram-negative bacterium and a pathogen for chronic periodontitis, has not been reported so far.

The three flavonoids (naringin, hesperidin, and rutin) are also found in citrus plants and have various bioactivities as potential therapeutic agents [22]. Naringin is found in grapefruits and has diverse therapeutic biological effects, such as anti-inflammatory and antioxidant effects [23,24,25]. Many studies have investigated the therapeutic effects of naringin in metabolic syndrome [26]. Prunin is a flavanone glucoside resulting from the hydrolysis of naringin. Hesperidin is a flavanone glycoside included in many citrus fruits at high concentrations [27] and exhibits anti-inflammatory, antioxidant, antitumor, and antimicrobial properties [28]. Rutin, the quercetin derivative, has antibacterial activity based on the inhibition of topoisomerase IV [29,30]. Transglycosylation is well known to significantly improve precursor compounds’ water solubility and bioavailability [31,32]. This also can mask their smell and taste [32,33,34,35]. There are no reports on the effects of lauroyl-esterified compounds of both these three flavonoids and their α-glucosyl derivatives against periodontal bacteria.

Therefore, this study evaluated these substances’ inhibitory effects in vitro on bacterial growth and biofilm formation. We also examined their cytotoxicity using human-derived cells and further assessed their inhibitory effects on periodontal disease in an animal model of periodontal pathogen infection. There is a demand for oral products with low-irritation substances that can be used by young children and the elderly. The substances from natural products in this study may serve as candidates for such products.

## 2. Materials and Methods

### 2.1. Bacterial Culture

All in vitro experiments used *P. gingivalis* 33277 (ATCC #33277). Animal experiments were performed using the *P. gingivalis* W83 strain provided by Dr. Kazuhisa Yamazaki of Niigata University Graduate School. *P. gingivalis* 33277 was cultured in tryptic soy broth (TSB) (Nissui Pharm, Tokyo, Japan) containing brain–heart infusion (B.D. Biosciences, Franklin Lakes, NJ, USA), 5 μg/mL hemin (Nacalai Tesque, Kyoto, Japan), and 1 μg/mL menadione (Nacalai tesque, Kyoto, Japan), or on the TSB agar plates including 1.5% agar. Hemin, which is not synthesized by *P. gingivalis* [36,37], and menadione, which is essential as the electron transport chain system of *P. gingivalis,* are required as the growth factor for *P. gingivalis* [38]. *P. gingivalis* W83 strains were cultured in the same TSB broth as *P. gingivalis* 33277, or on the TSB agar plate including 5% sheep defibrinated blood (NIPPON BIO-TEST LAB. Inc., Saitama, Japan). Both strains were incubated at 37 °C under anaerobic conditions using square jars and Anaeropak Kenki (Mitsubishi Gas Chemical, Tokyo, Japan).

### 2.2. Prunin Laurate and Its Analogues

In this study, we investigated seven different compounds. All flavonoids were purchased: prunin, naringin, hesperidin, and rutin from Merck (Darmstadt, Germany), and α-glucosyl flavonoids (α-glucosyl naringin, α-glucosyl hesperidin, and α-glucosyl rutin) from Toyo Sugar Refining Co., Ltd. (Tokyo, Japan). The preparation of prunin alkyl esters was previously described [19]. Briefly, a mixture in a screw-capped bottle containing 20 mL of acetone, 500 mg of molecular sieves for removing water from reaction mixtures, 0.04 mmol of flavonoid derivatives, 0.4 mmol of vinyl laurate, and 100 mg of the immobilized lipase Novozym 435 (Merck) was incubated at 50 °C for 72 h. Then, the immobilized enzyme and molecular sieves were removed by filtration. After evaporating the acetone under reduced pressure at 40 °C, the residual material was washed three times with hexane and then lyophilized. The purity was verified using high-performance liquid chromatography (HPLC), as described below. The reaction products were analyzed via HPLC using a Cosmosil 10C8-300 column (4.6 × 50 mm; Nacalai Tesque) at UV 280 nm. Elution was carried out in a linear gradient of 55% to 100% solvent A (methanol) for 4 min at a flow rate of 1.2 mL/min and 40 °C; 0.2% acetic acid solution was used as solvent B. We also employed structurally similar compounds to prunin, namely naringin, hesperidin, and rutin, esterified with lauric acid (Nar-C12, Hes-C12, and Rut-C12, respectively). Furthermore, α-glucosyl flavonoids (α-glucosyl naringin, α-glucosyl hesperidin, and α-glucosyl rutin) were used to enhance their water solubility, resulting in three additional compounds (αG-Nar-C12, αG-Hes-C12, and αG-Rut-C12, respectively). The HPLC results and the synthesized derivatives’ purities are shown in Appendix A. The structures of Pru-C12, Nar-C12, and αG-Nar-C12 are shown in Figure 1. As stock solutions, the above compounds were dissolved in dimethyl sulfoxide (DMSO) (Nacalai Tesque) at 50 mM.

### 2.3. Determination of Minimum Inhibitory Concentration (MIC)

*P. gingivalis* 33277 was grown anaerobically on TSB agar for 3–5 days and then suspended in TSB broth at an absorbance of 600 nm (Abs600) = 0.1 (2 × 10^8^ cfu/mL). The compounds (Pru-C12, Nar-C12, Hes-C12, Rut-C12, αG-Nar-C12, αG-Hes-C12, and αG-Rut-C12) were diluted in TSB broth to final concentrations of 100, 10, 1, 0.1, and 0.01 μM, respectively. A volume of 100 μL of the bacterial solution and each diluted compound were added to a 96-well plate. The plates were incubated at 37 °C for 48 h. After incubation, the absorbance at 630 nm (Abs_630_) was measured using a plate reader (680XR, BioRad, Hercules, CA, USA). The absorbance value of the culture medium was used as turbidity to determine bacterial growth and MIC [39].

### 2.4. Inhibition of Biofilm Formation

*P. gingivalis* 33277 with absorbance at 600 nm (Abs_600_) = 0.5 (1 × 10^9^ cfu/mL) was seeded in 24-well plates, and TSB broth containing 50 μM Pru-C12, Nar-C12, or αG-Nar-C12 was added. Biofilms were formed by incubation for 72 h under anaerobic conditions. After incubation, the biofilm was washed three times with PBS. Then, 0.5 mL of 2% crystal violet (Nacalai Tesque) solution was added to each biofilm and incubated at 37 °C for 1 min to stain the biofilm. After removing the staining solution and washing three times with PBS, 0.5 mL of 100% ethanol was added for decolorization. The absorbance of the decolorization solution was measured at 595 nm (680XR, Bio-Rad, Hercules, CA, USA), and the biofilm-forming ability in the presence of food additives was determined from the absorbance [40,41].

### 2.5. Cytotoxicity Test

HeLa cells, epithelial cells derived from human cervical cancer, were grown in DMEM (Nacalai Tesque, Kyoto, Japan) supplemented with 10% fetal bovine serum (FBS) (NICHIREI BIOSCIENCES Inc., Tokyo, Japan). The cells were seeded in 24-well plates at 100 µL each at 37 °C for 24 h under 5% CO_2_ at 37 °C. After incubation, Pru-C12, Nar-C12, and αG-Nar-C12 were added at final concentrations of 100, 10, 1, 0.1, and 0.01 μM and incubated for another 24 h. After incubation, 4 μL of MTT reagent (WST-8) (Nacalai Tesque) was added to each well, and the plates were incubated at 37 °C for 2 h under 5% CO_2_. Finally, the absorbance at 450 nm (Abs_450_) and reference wavelength 630 nm (Abs_630_) was measured using a plate reader. The cell viability was determined by the absorbance value [42].

### 2.6. Animal Experiments

A total of 20 six-week-old female Balb/c mice (Oriental Yeast Co., Ltd., Osaka, Japan) with an initial weight of 16 ± 2 g were used in the experiment. The diet was a powdered laboratory standard sample M.F. (Oriental Yeast Co., Ltd.). After a 3-day acclimatization period, the mice were administered water containing 1.0 mg/mL kanamycin (Nacalai Tesque) as an antimicrobial agent for 6 days to eliminate indigenous oral bacteria. Following the antimicrobial treatment, the mice were given regular water without antibiotics for 3 days to remove residual antimicrobials. Subsequently, the mice were divided into Pg-infection and control groups. The Pg-infection group received an oral application of *P. gingivalis* W83 strain at 1 × 10^10^ cfu/mL suspended in PBS with 2% carboxymethyl cellulose (CMC) (Merck KGaA, Darmstadt, Germany). The control group was treated with PBS containing 2% CMC. *P. gingivalis* W83 in PBS + 2% CMC and PBS + 2% CMC were topically applied every other day, 5 times, at a volume of 100 μL each time. From the beginning of the application, the mice were divided into four groups: one group receiving feed containing 0.02% Pru-C12 (Pru-C12-administered group), another group receiving regular feed (non-administered group), and two additional groups with or without the infection.

After the completion of the topical application, the mice were housed for 6 weeks and then sacrificed under anesthesia. Maxillary bones were collected, and heart blood was drawn. The extracted maxillary bones were stained with methylene blue (Nacalai Tesque) and observed under a stereo microscope (Olympus SZ-40, Tokyo, Japan). Images were captured, and the distance between the cementoenamel junction and alveolar bone crest (CEJ–ABC) on the buccal and palatal sides of the upper molars was measured using Image-J software ver. 1.53 (NIH, Bethesda, MD, USA).

The collected blood was centrifuged to obtain serum, and the concentration of IL-1β was measured using the Mouse IL-1 beta/IL-1F2 Immunoassay Quantikine ELISA Kit (R&D Systems, Minneapolis, MN, USA). The serum was stored frozen at −80 °C until use. The experimental procedure followed the protocol provided with the kit. Briefly, mouse serum or recombinant mouse IL-1β for the standard curve was added to a pre-coated 96-well plate with a monoclonal anti-mouse IL-1β antibody. After a 2-hour incubation, the plate was washed with PBS and horseradish peroxidase (HRP)-conjugated polyclonal anti-mouse IL-1β antibody was added followed by an additional 2-hour incubation. After washing with PBS, substrate tetramethylbenzidine (TMB) was added, and the reaction was stopped with diluted hydrochloric acid after a 30-minute incubation. The absorbances at 450 nm and 570 nm (Abs_450_, Abs_570_) were measured using a plate reader (680XR, Bio-rad).

The experimental procedures with animals implemented in this study were approved by the Osaka Prefecture University Animal Care and Committee (Animal Experiment No. 28-95). All procedures used in this study complied with the institutional policies of the Osaka Prefecture University Animal Care and Use Committee.

### 2.7. Statistical Analysis

The results of the MIC, biofilm formation inhibition, and cytotoxicity studies were analyzed via one-way analysis of variance, and multiple comparisons were made using the Dunnett method. One-way or two-way ANOVA was also used to analyze the alveolar bone resorption inhibition and blood IL-1β concentration, and multiple comparisons were made using the Tukey method. Prism9 (GraphPad Software ver. 9.5.1, Boston, MA, USA) statistical software was used for statistical analysis.

## 3. Results

### 3.1. Growth Inhibitory Effects of Prunin Laurates and Their Analogues

We conducted a growth inhibition test against *P. gingivalis* 33277 using seven compounds. Pru-C12 completely inhibited the growth of *P. gingivalis* at 10 μM (Figure 2A). Similarly, Nar-C12 and αG-Nar-C12 exhibited concentration-dependent growth inhibition of *P. gingivalis* starting from 10 μM (Figure 2B). Furthermore, Hes-C12, Rut-C12, αG-Hes-C12, and αG-Rut-C12 were found to significantly inhibit the growth of *P. gingivalis* at 100 μM (Figure 2C). Based on these findings, the minimum inhibitory concentration (MIC) was determined to be 10 μM for Pru-C12, Nar-C12, and αG-Nar-C12, while Hes-C12, Rut-C12, αG-Hes-C12, and αG-Rut-C12 exhibited an MIC of 100 μM or higher.

### 3.2. Biofilm Formation Inhibitory Effect of Prunin Laurate and Its Analogs

The biofilm formation inhibition test against *P. gingivalis* 33277 was conducted for three compounds, Pru-C12, Nar-C12, and αG-Nar-C12, which showed growth inhibitory effects. A concentration of 50 μM Pru-C12 significantly reduced *P. gingivalis* biofilm formation (Figure 3A). In addition, 50 μM Nar-C12 and αG-Nar-C12 also significantly reduced *P. gingivalis* biofilm formation (Figure 3B).

### 3.3. Cytotoxicity of Prunin Laurate and Its Analogs on Human-Derived Cells

To evaluate whether Pru-C12, Nar-C12, and αG-Nar-C12 exhibit cytotoxicity against human-derived cells at concentrations that affect the growth of *P. gingivalis*, we performed cytotoxicity tests using HeLa cells. Nar-C12 and αG-Nar-C12 were cytotoxic to HeLa cells at concentrations of 100 μM, while Pru-C12 was not cytotoxic at the concentrations tested in this study (Figure 4).

### 3.4. Investigation of Periodontal Disease Prevention Effect of Pru-C12 in a Mouse Animal Model of Periodontal Disease Infection

We conducted animal experiments using an animal model of periodontal disease infection in mice using Pru-C12, which did not show cytotoxicity against HeLa cells at the concentrations tested in this study. The CEJ–ABC distance on the buccal and palatal sides of the right molar of the maxilla was measured. The CEJ–ABC distance increased with *P. gingivalis* infection on both buccal and palatal sides, suggesting the induction of alveolar bone resorption. On the other hand, in the Pru-C12-treated group, the increase in CEJ–ABC distance due to infection with *P. gingivalis* W83 strain tended to decrease both buccally and palatally compared to the non-treated group (Figure 5).

In addition, measurement of the concentration of the inflammatory cytokines IL-1β and TNF-α in the blood showed that *P. gingivalis* W83 infection caused an increasing trend in IL-1β concentration compared to non-infection, but no change in TNF-α concentration. Pru-C12 administration did not affect IL-1β concentration (Figure 6).

## 4. Discussion

In this study, we investigated the effects of several flavanone and lauric acid esters and their derivatives on the growth and biofilm formation of *P. gingivalis*, a pathogenic bacterium of chronic periodontitis, to search for naturally occurring hypoallergenic substances that inhibit the progression of periodontal disease. Cytotoxicity against human cells was also evaluated at the in vitro level. Then, the local and systemic effects of Pru-C12, considered most beneficial based on in vitro experimental results, were assessed in vivo in an animal model of periodontal disease infection in mice. As a result, all seven samples examined showed growth inhibitory effects on *P. gingivalis* (Figure 2). In addition, Pru-C12, Nar-C12, and αG-Nar-C12 inhibited *P. gingivalis* biofilm formation at concentrations that inhibited growth (Figure 3). Nar-C12 and αG-Nar-C12 showed cytotoxicity against HeLa cells at 100 μM, while Pru-C12 showed no cytotoxicity at 100 μM (Figure 4). In animal experiments, experimental alveolar bone resorption caused by *P. gingivalis* infection was observed, indicating that Pru-C12 administration inhibited alveolar bone resorption (Figure 5). Examination of serum samples showed that *P. gingivalis* infection tended to increase IL-1β levels in the blood, but Pru-C12 did not suppress IL-1β levels (Figure 6).

It is well-known that keeping oral bacteria below a certain amount prevents or controls periodontal disease [43,44,45]. Therefore, it is not always necessary to completely sterilize periodontal pathogenic bacteria. However, many oral hygiene products currently in use contain synthetic chemicals and alcohol, highly irritating antimicrobial agents intended for disinfection. Bacteria attempt to survive exposure to antimicrobials in various ways, which can result in the development of drug-resistant bacteria [46]. In other words, a disinfectant action that is too decisive can create new oral hygiene problems. In addition to the risk of drug-resistant bacteria, there are other problems. For example, ethanol in mouthwashes and types of liquid toothpastes is highly irritating when placed in the mouth, making it unsuitable for older people, the primary sufferers of periodontal disease, and young children. The inhibitory concentration of Pru-C12, Nar-C12, and αG-Nar-C12 on *P. gingivalis* was 10 μM, which is 10 to 300 times lower than that of common antimicrobial agents [47,48], so the possibility of bacterial resistance emerging is considered low.

On the other hand, the water solubility of αG-Nar-C12 has been improved. On the other hand, the results of αG-Nar-C12, αG-Hes-C12, and αG-Rut-C12, which are analogs to which sugar was added in the expectation that the growth inhibitory effect would also increase by improving water solubility, were not significantly different from those of Nar-C12, Hes-C12, and Rut-C12 without addition. Therefore, it can be said that the number of sugar chains does not affect the growth inhibitory effect as far as the results of this experiment are concerned. In addition, because Pru-C12 is tasteless and some naringin derivatives showed less irritation to the skin and eyes in rabbits [49], substances derived from natural substances of food origin may show low irritation in the oral cavity as ingredients in oral care products. However, whether or not Pru-C12 has low irritation will be analyzed in the future.

Since periodontopathogenic bacteria exist in the oral cavity, it is considered necessary for Pru-C12 to act rapidly in the oral cavity. Although we could not determine the minimal bactericidal concentrations (MBICs) for the low solubility of most derivatives in this study, the time-kill assay [50] was performed for Pru-C12, and 20 μg/mL Pru-C12 reduced the number of viable *P. gingivalis* bacteria to 0 after 2 h of incubation (Appendix A). Therefore, Pru-C12 may be helpful in the prevention of periodontal disease when added to gel oral care products that can remain in the mouth for an extended period. Next, we attempted to determine each compound’s minimum bactericidal concentration. Still, except for Pru-C12, the MBIC could not be determined because the compound precipitated in the *P. gingivalis* culture medium at higher concentrations than those defined by the MIC. A time-kill assay was performed at 20 µg/mL (33 µM) against *P. gingivalis* and confirmed that viable *P. gingivalis* bacteria were no longer detected after 2h of incubation mixed with Pru-C12 (Appendix A).

Oral biofilm, or dental plaque, is an accumulation of various substances (proteins, polysaccharides, etc.) secreted by the host and oral bacteria and is mainly formed on tooth surfaces and root surfaces in the oral cavity [3,51]. In addition, periodontopathogenic bacteria are predominantly anaerobic and prefer to live in shallow oxygen conditions such as periodontal pockets [51]. Bacteria in biofilms are thought to be less susceptible to antimicrobial agents because their extracellular matrix prevents penetration into the biofilm, and antibiotics require concentrations 100 to 500 times higher than those used against single bacteria [52]. Therefore, it is necessary to inhibit the growth of bacteria. Consequently, it is crucial not only to inhibit bacterial growth but also to inhibit biofilm formation. This study showed that Pru-C12, Nar-C12, and αG-Nar-C12 inhibited bacterial growth and biofilm formation at concentrations of 50 μM, which is more valuable than substances effective only on the bacteria or biofilms. Although Pru-C12, Nar-C12, and αG-Nar-C12 inhibited *P. gingivalis* biofilm formation in this study, whether they have bacteriostatic or bactericidal effects on the bacteria in the biofilm itself has not been investigated.

The gingival epithelium is the first site in the oral cavity where periodontopathogenic bacteria meet the host, and thus plays a vital role in the pathogenesis of periodontal disease [53]. The epithelium consists of epithelial cells, which act as a physical barrier against bacterial infection and a defense mechanism to detect foreign microbes and foreign substances, producing antimicrobial substances and inflammatory cytokines. Therefore, the components in oral care products should not exhibit cytotoxicity to epithelial cells. This study showed that Pru-C12 was not cytotoxic to HeLa cells derived from human cervical carcinoma epithelial cells, even at 100 μM, and Pru-C12 inhibited the bacterial growth of *P. gingivalis* at 10 μM. Pru-C12 is not cytotoxic to HeLa cells, even at concentrations 10-fold higher than those that inhibit bacterial growth. In this respect, Pru-C12 is superior to Nar-C12 and αG-Nar-C12, which showed similar inhibitory effects against *P. gingivalis*. Céliz et al. [54] reported the cytotoxicity of prunin- and hesperetin glucoside-alkyl (C4–C18) esters against Jurkat cells. Pru-C12 shows no difference in cytotoxicity compared to hesperetin glucoside-C12 and similar flavonoid ester derivatives, but Pru-C12 shows less cell lysis activity than hesperetin glucoside-C12. Therefore, Pru-C12 may have less cytolytic activity against animal cell membranes due to its chemical structure than similar compounds in this study.

Next, we evaluated the periodontal disease inhibitory effect of Pru-C12, which was most promising in the commonly used mouse periodontal infection model [55,56] at the test tube level. The alveolar bone resorption was compared using the CEJ–ABC distance as a marker, and the increase in CEJ–ABC induced by *P. gingivalis* infection was not significantly different after treatment with Pru-C12. Still, there was a trend toward a decrease in CEJ–ABC. Pru-C12 can potentially suppress the progression of periodontal disease in vivo. In the present experiment, the dose of Pru-C12 was estimated to be 0.02% of the diet, or 40 mg/kg⋅day per mouse, which may not be sufficient to suppress alveolar bone resorption significantly [23,24,25,26]. Still, at the same time, it is predicted that increasing the dosage may produce promising results.

Since periodontal disease has recently been suggested to be associated with systemic diseases [5,6,7,8,9,10,11,12], heart blood samples were taken from mice used in the experiment, and the concentrations of IL-1β and TNF-α were measured, which are included as inflammatory cytokines in the blood serum. However, administration of Pru-C12 did not decrease the increasing trend of IL-1β concentration in the blood. TNF-α showed no change between no infection and infection. Therefore, assuming periodontal disease in humans, it is necessary to extend the period of infection, etc., to evaluate the systemic effects of periodontal disease. The effect of Pru-C12 on the suppression of systemic inflammation could not be evaluated in this study and is a future issue.

This study showed that Pru-C12 and its analogs inhibit bacterial growth of *P. gingivalis,* but the mechanism of the growth inhibition is not yet known. Growth inhibition by Pru-C12 has four main possible mechanisms: the destruction of the bacterial membrane, the inhibition of DNA or RNA synthesis, the inhibition of protein synthesis, and the inhibition of metabolism. Céliz et al. reported 150 μg/mL (nearby 243 μM) of not prunin but Pru-C12 showed an inhibitory effect against several Gram-negative and Gram-positive bacteria species [20]. Thus, a lauroyl chain moiety in Pru-C12 is essential for growth inhibition and could be associated with the bacterial membrane. However, the researchers also examined the mode of action for growth inhibition of *L. monocytogenes*, one of the food-poisoning Gram-positive bacteria, using scanning transmission electron microscopy. They revealed that the bacterial membrane was not lytic [20]. Our preliminary nucleic acid and protein elution tests from *P. gingivalis* bacteria showed that 10 μM Pru-C12 did not elute even 2 h after addition in our preliminary experimental results, suggesting that physical damage to the cell membrane is not the cause of growth inhibition. Other possible inhibition mechanisms include inhibition of DNA or RNA and protein synthesis and metabolism, which will be investigated in the future.

Periodontal disease should be prevented at all stages of life [44,45]. However, many oral hygiene products on the market are highly irritating and intended for sterilization, making them difficult for use by the elderly and young children. This study found that Pru-C12 and its analogs derived from natural substances inhibited periodontal disease in vitro and in vivo. Thus, Pru-C12 is a candidate ingredient of natural origin with low irritation for oral care products. These findings may be helpful for the development of oral hygiene products that are easy to use for people at all stages of life, including the elderly and young children.

## Figures and Tables

**Figure 1 foods-13-01917-f001:**
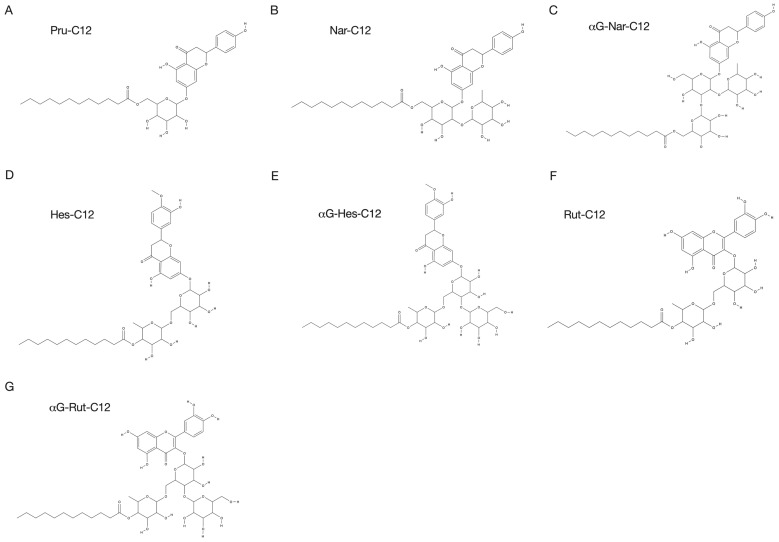
Prunin laurate and its analogs. The structural formulas of Pru-C12 (**A**), Nar-C12 (**B**), αG-Nar-C12 (**C**), Hes-C12 (**D**), αG-Hes-C12 (**E**), Rut-C12 (**F**), and αG-Rut-C12 (**G**). The structures of (**D**,**E**,**G**) are predicted.

**Figure 2 foods-13-01917-f002:**
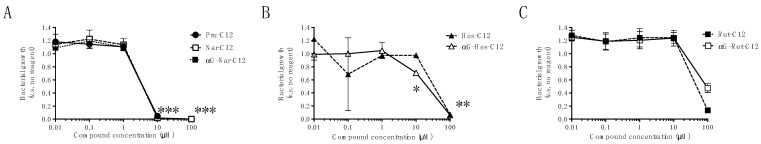
Effect of Pru-C12, Nar-C12, αG-Nar-C12, Hes-C12, Rut-C12, αG-Hes-C12, and αG-Rut-C12 on the growth of *P. gingivalis*. (**A**): Pru-C12, Nar-C12, and αG-Nar-C12; (**B**): Hes-C12 and αG-Hes-C12; and (**C**): Rut-C12 and αG-Rut-C12. Data in the graphs represent the mean ± S.D. (*n* = 3). Representative data for two or more repeated experiments are shown. One-way analysis of variance was used for statistical analysis, and the Dunnett method was used for multiple comparisons. *: *p* < 0.05 indicates significance relative to the bacterial growth rate in the absence of each sample.

**Figure 3 foods-13-01917-f003:**
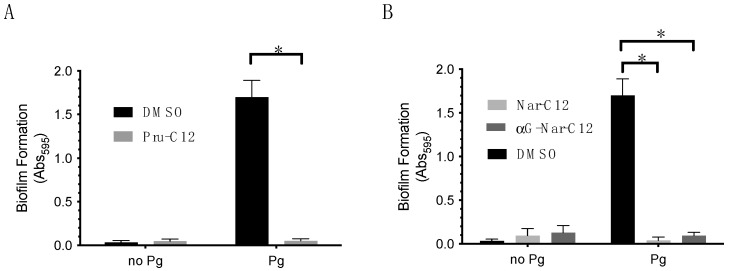
Effects of Pru-C12 (**A**), Nar-C12, and αG-Nar-C12 (**B**) on *P. gingivalis* biofilm formation. The graphs represent the mean ± S.D. (*n* = 3) and show representative data for two or more repeated experiments. *p* < 0.05 for *, indicating significance relative to the amount of biofilm formation in the absence of Pru-C12, Nar-C12, and αG-Nar-C12. One-way analysis of variance was used for statistical analysis. *: *p* < 0.05 indicates significance relative to the bacterial growth rate in the absence of each sample.

**Figure 4 foods-13-01917-f004:**
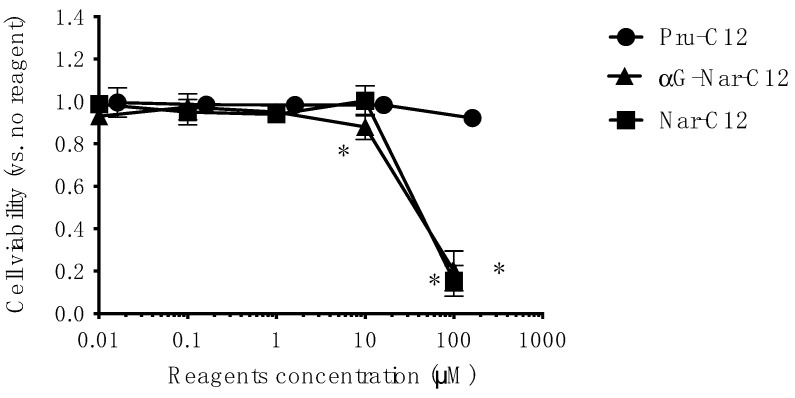
Cytotoxic effects of Pru-C12, Nar-C12, and αG-Nar-C12 on human-derived cells. The cytotoxicity against HeLa cells. The graphs represent mean ± S.D. (*n* = 4) and show representative data from two or more experiments. *: *p* < 0.05 indicates significance for cell viability in the absence of Pru-C12, Nar-C12, or αG-Nar-C12. One-way analysis of variance was used for statistical analysis, and the Dunnett method was used for multiple comparisons.

**Figure 5 foods-13-01917-f005:**
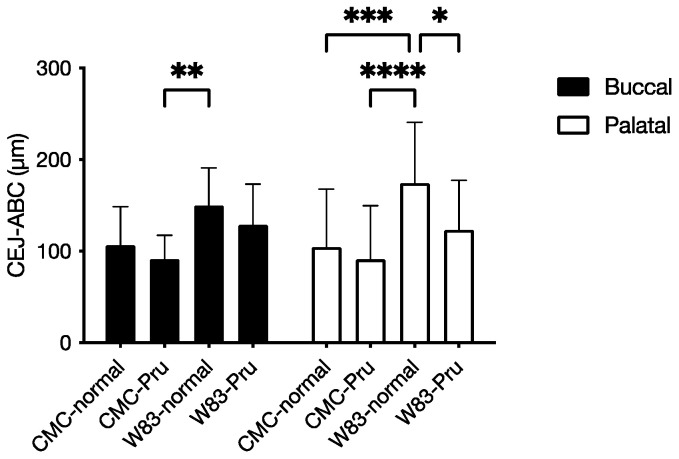
Inhibition of alveolar bone resorption by Pru-C12 in the maxilla of an animal model of periodontal disease in mice. Pru-C12 was administered at the same time as infection with *P. gingivalis* W83; the mouse was slaughtered, the maxillary bone was extracted, stained with methylene blue, and the CEJ–ABC distance was measured. Each bar in the graph represents mean ± S.D. *n* = 4, data from one experiment. * indicates *p* < 0.05, indicating that the CEJ–ABC distance during *P. gingivalis* W83 infection is significant relative to the alveolar bone resorption during no infection. Two-way analysis of variance was used for statistical analysis, and the Tukey method was used for multiple comparisons.

**Figure 6 foods-13-01917-f006:**
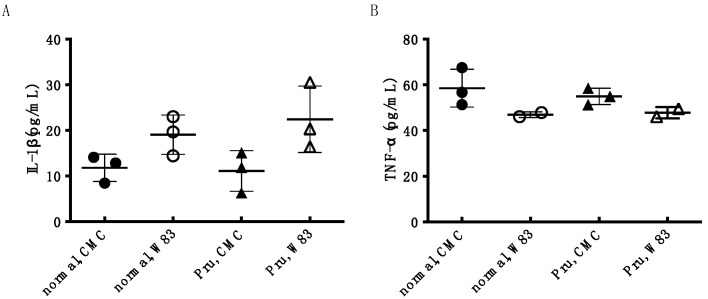
Inhibitory effect of Pru-C12 on IL-1β (**A**) and TNF-α (**B**) concentrations in an animal model of mouse periodontal disease. Mouse serum samples were added to 96-well plates coated with anti-mouse IL-1β or TNF-α antibody, and after reaction and washing with PBS, HRP-conjugated polyclonal anti-mouse IgG antibody was added, and the reaction was carried out for another 2 h. After washing with PBS, substrate TMB was added, and the reaction was stopped by adding dilute hydrochloric acid. The absorbances at 450 nm and 570 nm (Abs450, Abs570) were measured with a plate reader. Each data point in the graph shows the mean ± S.D. *n* = 3, data from one experiment. One-way analysis of variance was used for statistical analysis, and Tukey’s method was used for multiple comparisons.

## Data Availability

The original contributions presented in the study are included in the article/Appendix A, further inquiries can be directed to the corresponding author.

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
