# Peer review of "Prunin Laurate Derived from Natural Substances Shows Antibacterial Activity against the Periodontal Pathogen Porphyromonas gingivalis"

_foods, 2024, doi:10.3390/foods13121917_

Round 1

Reviewer 1 Report

Comments and Suggestions for Authors

The article by E. Wada et al. describes the biological activities of 7 flavonoid esters of fatty acid lauric acid against periodontitis-related bacteria and their ability of forming biofilms. The topic is very interesting given the high prevalence of periodontal disease and the connection with other diseases.

The article provides biological data in vitro and in vivo (mice) that show the effectiveness of prunine laurate against P. gingivalis while exhibiting low cytotoxicity (HeLa cells). The authors suggest its use in food supplements and in oral products for the prevention and management of periodontitis.

I recommend it in Foods after addressing the following issues:

1)      Line 83: “Investigating the effects of these low-irritation substances derived from natural products“. Only because these compounds have a natural origin, it does not guarantee that they are not irritating. Experiments or references from literature must be given to support this. Again it is mentioned in lines 283 and 326-327 and 395.

2)      Line 106-107: The structures of ”hesperidin and rutin“ should be given in the Figure 1. Moreover, some more information and references about these compounds and naringin, could be included in the introduction (line 81).

3)      Lines 107-108: it is not mentioned how the derivatives with additional glucose were prepared. Why the derivative aG-Pru-C12 was not prepared?

4)      Lines 301 and 356-357: The low cytotoxicity of Pru-C12 versus the other similar compounds is not explained, although it is important discovery. Are there other studies in the literature to support the result?

Comments on the Quality of English Language

No significant errors were found.

Author Response

Reply to the reviewer’s comments

Response to the Reviewer 1

Thank you for your comments and suggestions. Point-to-point responses are as follows. Changes in the text are highlighted in yellow.

  1. Line 83: “Investigating the effects of these low-irritation substances derived from natural products.” Only because these compounds have a natural origin, it does not guarantee that they are not irritating. Experiments or references from literature must be given to support this. Again it is mentioned in lines 283 and 326-327 and 395.

Authors’ response: We agree with the reviewer’s comment that the substances from natural products are not always low irritation. On the other hand, there are no reports of irritation for prunin, but some naringin derivatives showed less irritation to the skin and eyes in rabbits (Lather, Sharma, and Khatkar 2020). Thus, following the reviewer's advice, the sentences (lines 83, 326-327, and 393) were removed and rewritten in the introduction and discussion sections (lines 71 - 72, 99 - 101, 349 - 353, and 439 - 442). Moreover, irritability fell outside the scope of this experiment, so it has been designated as a future research task.

Lather A, Sharma S, Khatkar A. Naringin derivatives as glucosamine-6-phosphate synthase inhibitors based preservatives and their biological evaluation. Sci Rep 2020;10:20477.

Lines 71 - 72

The substances derived from natural products may potentially exhibit lower irritability compared to chemically synthesized substances.

Lines 99 - 101.

There is a demand for oral products that can be used by young children and the elderly who utilize low-irritation substances. The substances from natural products in this study may serve as candidates for such products.

Lines 349 - 353.

In addition, because Pru-C12 is tasteless (data not shown) and some naringin derivatives showed less irritation to the skin and eyes in rabbits [46], the substances derived from natural substances of food origin may show low-irritative in the oral cavity as an ingredient of oral care products. However, whether or not Pru-C12 has low irritation will be analyzed in the future.

Lines 439 - 442.

Thus, Pru-C12 is a candidate ingredient of natural origin with low irritation for oral care products. These findings may be helpful for the development of oral hygiene products that are easy to use for people at all stages of life, including the elderly and young children.

  1. Line 106-107: The structures of “hesperidin and rutin” should be given in the Figure 1. Moreover, some more information and references about these compounds and naringin, could be included in the introduction (line 81).

Authors’ response: Following the reviewer's advice, we have provided the structural formulas for the seven compounds, which HPLC confirmed to be >85% purified (Fig. S1) and used in our experiments. Still, the structures of Hes-C12, aG-Rut-C12, and aG-Hes-C12 were not confirmed by NMR. Thus, the predicted structural formulas for these three compounds are shown in Figure 1. Information and references for naringin, hesperidin, and rutin have been added in the introduction (lines 83 - 95).

Lines 83 - 95.

The three flavonoids (naringin, hesperidin, and rutin) are also found in citrus plants and have various bioactivities as potential therapeutic agents [22]. Naringin is found in grapefruits and has diverse therapeutic biological effects, such as anti-inflammatory and antioxidant effects [23–25]. Many studies have investigated the therapeutic effects of naringin in metabolic syndrome [26]. Prunin is a flavanone glucoside resulting from the hydrolysis of naringin. Hesperidin is a flavanone glycoside, also included in many citrus fruits with high concentrations [27]and exhibits anti-inflammatory, antioxidant, antitumor, and antimicrobial properties [28]. Rutin, the quercetin derivative, has antibacterial activity based on the inhibition of topoisomerase IV [29,30].Transglycosylation is well known to significantly improve precursor compounds’ water solubility and bioavailability [31,32]. It also can mask its smell and taste [32–35]. There are no reports on the effects of lauroyl-esterified compounds of both these three flavonoids and their α-glucosyl derivatives against periodontal bacteria.

  1. Lines 107-108: it is not mentioned how the derivatives with additional glucose were prepared. Why the derivative ɑG-Pru-C12 was not prepared?

Authors’ response: The derivatives with additional glucose were synthesized by esterification of vinyl ester to ɑ-glucosyl flavonoids except ɑ-glucosyl-prunin purchased from the Toyo Sugar Refining Co. Ltd. ɑ-glucosyl prunin was not commercially available, and prunin is a flavanone glucoside resulting from the hydrolysis of naringin. To obtain more hydrophobicity, we used ɑ-glucosyl flavonoids for synthesizing these derivatives. Thus, we did not prepare ɑG-Pru-C12. Hence, we added the sentences for preparing these flavonoid esters and a supplemental figure (Fig. S1) indicating the purity of synthesized compounds in the Material and Methods section (lines 115 - 129, 132 - 133, and 137 - 138).

Lines 115 - 129.

All flavonoids were purchased: prunin, naringin, hesperidin, rutin from Merck (Darmstadt, Germany), and ɑ-glucosyl flavonoids (ɑ-glucosyl naringin, ɑ-glucosyl hesperidin, ɑ-glucosyl rutin) from Toyo Sugar Refining Co. Ltd. (Tokyo, Japan). The preparation of prunin alkyl esters was previously described [19]. Briefly, a mixture containing 20 mL of acetone, 500 mg of molecular sieves, 0.04 mmol flavonoid derivatives, 0.4 mmol vinyl laurate, and 100 mg of the immobilized lipase Novozym 435 (Merck) in a screw-capped bottle was incubated at 50 °C for 72 h. Then, the immobilized enzyme and molecular sieves were removed by filtration. After evaporating the acetone under reduced pressure at 40 °C, the residual material was washed three times with hexane and then lyophilized. Purity was verified by high-performance liquid chromatography (HPLC) as described below. Reaction products were analyzed by HPLC using a Cosmosil 10C8-300 column (4.6 × 50 mm; Nacalai Tesque) at UV 280 nm. Elution was carried out in a linear gradient from 55% to 100% solvent A (methanol) for 4 min at a flow rate of 1.2 mL/min and 40 °C; 0.2% acetic acid solution was used as solvent B.

Lines 132 - 133.

The HPLC results and the synthesized derivatives’ purity are shown in Fig. S1.

Lines 137 - 138

Hes-C12 (D), αG-Hes-C12 (E), Rut-C12 (F), and αG-Rut-C12 (G). The structures of D, E, and G are predicted.

  1. Lines 301 and 356-357: The low cytotoxicity of Pru-C12 versus the other similar compounds is not explained, although it is important discovery. Are there other studies in the literature to support the result?

Authors’ response: Although cytotoxicity of prunin- and hesperetin- glucoside-alkyl (C4-C18) esters against Jurkat cells has been reported (Céliz et al. 2013), there is no difference in cytotoxicity between Pru-C12 and hesperetin glucoside-C12 However, in the same paper, the authors found no difference in cytotoxicity between Pru-C12 and hesperetin glucoside-C12, with potent cytotoxicity at 100 µM. However, the same paper examines cell lysis and shows that hesperetin glucoside-C12 is less cytolytic than Pru-C12, even at 100 µM. Therefore, Pru-C12 may have less cytolytic activity against animal cell membranes due to its chemical structure than similar compounds in this study. Thus, the following text and references have been added to the Discussion section (lines 394 - 399).

Lines 394 - 399

Céliz et al. [51]reported the cytotoxicity of Prunin- and hesperetin glucoside-alkyl (C4-C18) esters against Jurkat cells. Pru-C12 shows no difference in cytotoxicity compared to hesperetin glucoside-C12, similar flavonoid ester derivatives, but Pru-C12 shows less cell lysis activity than hesperetin glucoside-C12. Therefore, Pru-C12 may have less cytolytic activity against animal cell membranes due to its chemical structure than similar compounds in this study.

Céliz G, Alfaro FF, Cappellini C et al. Prunin- and hesperetin glucoside-alkyl (C4–C18) esters interaction with Jurkat cells plasma membrane: Consequences on membrane physical properties and antioxidant capacity. Food Chem Toxicol 2013;55:411–23.

Reviewer 2 Report

Comments and Suggestions for Authors

The manuscript entitledPrunine laurate derived from natural substances shows antibac-  terial activity against the periodontal pathogen Porphyromonas  gingivalis” by  Erika Wada et al ., describes the antibacterial effects of prunine laurate and its analogs  on periodontopathogenic bacteria: Below are some examples of issues on which the authors might want to work on.

1.      Authors should perform minimum bactericidal concentrations (MBCs) which will show the lowest concentration of Prunine laurate and their analogs against P. gingivalis. Further both the results of MIC and MBC should be presented in the table form so that it will be clear for readers.  

2.      For the biofilm assay quantitative is not enough to show the effect of biofilm by Prunine laurate. However, authors should perform confocal fluorescence imaging to evaluate the effect of Prunine laurate and their analogs on biofilm from this assay. From this assay authors can  visualize and quantify biofilm formation. (https://pubmed.ncbi.nlm.nih.gov/9777582/)

3.      Cytotoxicity assay should be performed with all the prunine laurate and its analogs. Why do authors use Hela cells why not HEK 293 cells.   

4.      Results and discussion should be combined and more emphasis should be on the mechanism of Prunine laurate on P. gingivalis.

The manuscript is not acceptable in the present format.

Comments on the Quality of English Language

english is fine 

Author Response

Response to the Reviewer 2

Thank you for your comments and suggestions. Point-to-point responses are as follows. Changes in the text are highlighted in yellow.

  1. Authors should perform minimum bactericidal concentrations (MBCs) which will show the lowest concentration of Prunin laurate and their analogs against gingivalis. Further, both the results of MIC and MBC should be presented in the table form so that it will be clear for readers.

Authors’ response: In general, MBIC indicates higher concentrations than MIC. Because of their low water solubility, the acyl flavonoid derivatives studied in this study could not examine MBIC at higher concentrations than the MIC test. However, Pru-C12, which has relatively high water solubility, was shown to kill P. gingivalis after incubation at 20 µg/mL (33 µM) for 2 hours. Thus, we added the sentences (lines 355 - 358 and 360 - 366) in the discussion section with the supplemental document (Supplementary method and Fig. S2).

Lines 355 - 358.

Although we could not determine the minimal bactericidal concentrations (MBICs) for the low-solubility of most derivatives in this study, the Time-Kill Assay [47] was performed for Pru-C12, and 20 μg/mL Pru-C12 reduced the number of viable P. gingivalis bacteria to 0 after 2 hours of incubation (Fig. S2).

Lines 360 - 366.

Next, we attempted to determine each compound’s minimum bactericidal concentration (MBIC). Still, except Pru-C12, MBIC could not be determined because the compound precipitated in the P. gingivalis culture medium at higher concentrations than those defined by the MIC. Time kill assay was performed at 20 µg/mL (33 µM) against P. gingivalis and confirmed that P. gingivalis viable bacteria were no longer detected after 2h of incubation mixed with Pru -C12 (Fig. S1 and Supplementary method).

  1. For the biofilm assay quantitative is not enough to show the effect of biofilm by Prunin laurate. However, authors should perform confocal fluorescence imaging to evaluate the effect of Prunin laurate and their analogs on biofilm from this assay. From this assay authors can visualize and quantify biofilm formation. (https://pubmed.ncbi.nlm.nih.gov/9777582/)

Authors’ response: We, unfortunately, do not have a confocal microscope. This method in the present study has also been used in many papers, and we believe it is acceptable as an experiment for inhibition of biofilm formation.

  1. Cytotoxicity assay should be performed with all the prunin laurate and its analogs. Why do authors use Hela cells why not HEK 293 cells.

Authors’ response: Since we planned to select the most active candidate compounds for animal studies, we examined only those active at low concentrations from the MIC results. Since there have been many recent reports examining cytotoxicity using HeLa cells, as in the references listed below (Choudhuri et al. 2020; Millones-Gómez et al. 2022; Shi et al. 2022; Hoang, Alfarraj and Alharbi 2024), we used Hela cells instead of HEK 293 cells in this study.

Choudhuri I, Khanra K, Pariya P et al. Structural Characterization of an Exopolysaccharide Isolated from Enterococcus faecalis, and Study on its Antioxidant Activity, and Cytotoxicity Against HeLa Cells. Curr Microbiol 2020;77:3125–35.

Hoang T-V, Alfarraj S, Alharbi SA. An investigation on antimicrobial and anticancer competence of macro red algae under in-vitro condition. Environ Res2024:119026.

Millones-Gómez PA, Garza-Ramos MAD la, Urrutia-Baca VH et al. Cytotoxicity of Peruvian propolis and Psidium guajava on human gingival fibroblasts, PBMCs and HeLa cells. F1000Research 2022;11:430.

Shi P, Liu Z, Cen R et al. Three new compounds from the dried root bark of Wikstroemia indica and their cytotoxicity against HeLa cells. Nat Prod Res2022;36:5476–83.

  1. Results and discussion should be combined and more emphasis should be on the mechanism of Prunin laurate on gingivalis.

Authors’ response: As the reviewer suggested, we also considered combining the results and discussion into one, but we did not combine them because we wanted to discuss the experimental results carefully one by one. Growth inhibition by Pru-C12 has three main possible mechanisms: the destruction of the bacterial membrane, inhibition of DNA or RNA synthesis, and inhibition of protein synthesis. We are conducting experiments on them but have not obtained precise data to determine which causes the growth inhibition. We consider clarifying the mechanism of the antimicrobial activity in this case to be a future issue. However, we added the sentences discussed by citing the previous report in the Discussion section (lines 379 - 382 and 419 - 429).

Lines 419 - 429.

This study showed that Pru-C12 and its analogs inhibit bacterial growth of P. gingivalis, but the mechanism of the growth inhibition is not yet known. Growth inhibition by Pru-C12 has four main possible mechanisms: the destruction of the bacterial membrane, the inhibition of DNA or RNA synthesis, the inhibition of protein synthesis, and the inhibition of metabolism. Céliz et al. reported 150 μg/ml (nearby 243 μM) of not prunin, but Pru-C12 showed an inhibitory effect against several Gram-negative and Gram-positive bacteria species [20]. Thus, a moiety of lauroyl chain in Pru-C12 is essential for growth inhibition and could be associated with the bacterial membrane. However, they also examined the mode of action for growth inhibition of L. monocytogenes, one of the food-poisoning Gram-positive bacteria, using scanning and transmission electronic microscopies. They revealed that the bacterial membrane was not lytic [20].

Lines 379 - 382.

Although Pru-C12, Nar-C12, and αG-Nar-C12 inhibit P. gingivalis biofilm formation in this study, whether they have bacteriostatic or bactericidal effects on the bacteria in the biofilm itself has not been investigated.

Reviewer 3 Report

Comments and Suggestions for Authors

General comments

This manuscript evaluates the antimicrobial and antibiofilm activity of Prunin laurate derived from natural substances against the periodontal pathogen Porphyromonas  gingivalis.  The study is of interest to the field of odontology. The experimental work is in general well done. Results are clearly commented. However some clarifications are necessary.

 Specific comments

 Materials and methods

 Which is the culture collection of P. gingivalis 33277?

 Add the function of hemin and menadione in the medium (see line 108)?

 Explain the use of 500 mg of molecular sieve /line 119).

 Rewrite the  sentence about the use of glucose to enhance solubility.

 Results

Avoid the inclusion of information about Materials and Methods. For example, the info mentioned at lines: 223-226, 244-248, 262-263, 277-281.

Comments on the Quality of English Language

no comments

Author Response

Thank you for your comments and suggestions. Point-to-point responses are as follows. Changes in the text are highlighted in yellow.

Materials and methods

  1. Which is the culture collection of gingivalis 33277?

Authors’ response: P. gingivalis 33277 came from ATCC collection (ATCC #33277). We added the catalog number of the ATCC collection (lines 104) in the Materials and Methods section.

Line 104

All in vitro experiments used P. gingivalis 33277 (ATCC #33277).

  1. Add the function of hemin and menadione in the medium (see line 108)?

Authors’ response: P. gingivalis cannot synthesize heme [Fe(II)‐protoporphyrin IX] or hemin [Fe(III)‐protoporphyrin IX‐Cl], which are important growth/virulence factors (Gibbons et al. 1960; Shah et al. 1979; Schifferle et al. 1996; Olczak et al. 2005). Menadione is a synthetic Vitamin K, called K3, and is required for P. gingivalis growth (Gibbons et al. 1960) because P. gingivalis has a menaquinone with nine isoprenyl units (MK-9), and menadione is thought to be converted to MK-9 in P. gingivalis cells. Menaquinone participates in bacteria's electron transport chain systems with anaerobic respiratory chain systems, including P. gingivalis (Saiki et al. 2023). Thus, we added the sentence in the Materials and Methods section (lines 109 - 112).

Gibbons RJ, Macdonald JB. Hemin and vitamin K compounds as required factors for the cultivation of certain strains of Bacteroides melaninogenicus. J Bacteriol. 1960;80:164–170.

Shah HN, Bonnett R, Matten B, Williams RA. The porphyrin pigmentation of subspecies of Bacteroides melaninogenicus. Biochem J. 1979;180:45–50.

Schifferle RE, Shostad SA, Bayers‐Thering MT, Dyer DW, Neiders ME. Effect of protoporphyrin IX limitation on Porphyromonas gingivalis. J Endod. 1996;22:352–355.

Olczak T, Simpson W, Liu X, Genco CA. Iron and heme utilization in Porphyromonas gingivalis. FEMS Microbiol Rev. 2005;29:119–144.

Saiki K, Urano-Tashiro Y, Yamanaka Y, Takahashi Y. Phylloquinone is preferable over menadione as a growth factor for Porphyromonas gingivalis. J Oral Biosci. 2023 Dec;65(4):273-279.

Lines 109 - 112

Hemin, which is not synthesized by P. gingivalis [36,37], and menadione, which is essential as the electron transport chain system of P. gingivalis, are required as the growth factor for P. gingivalis [38].

  1. Explain the use of 500 mg of molecular sieve /line 119).

Authors’ response: Adding molecular sieves can remove water from reaction mixtures, thereby inhibiting the hydrolysis reaction and promoting the ester transfer reaction by lipases. Thus, we added the words for explanation (line 122).

Line 122

Briefly, a mixture containing 20 mL of acetone, 500 mg of molecular sieves for removing water from reaction mixtures, 0.04 mmol flavonoid derivatives, 0.4 mmol vinyl laurate, and 100 mg of the immobilized lipase Novozym 435 (Merck) in a screw-capped bottle was incubated at 50 °C for 72 h.

  1. Rewrite the sentence about the use of glucose to enhance solubility.

Authors’ response: Following the reviewer’s suggestion, we rewrote the sentences in the Materials and Methods section (Lines 133 - 135).

Lines 133 - 135

Furthermore, ɑ-glucosyl flavonoids (ɑ-glucosyl naringin, ɑ-glucosyl hesperidin, ɑ-glucosyl rutin) were used to enhance their water solubility, resulting in three additional compounds (αG-Nar-C12, αG-Hes-C12, αG-Rut-C12).

Results

  1. Avoid the inclusion of information about Materials and Methods. For example, the info mentioned at lines: 223- 226, 244-248, 262-263, 277-281.

Authors’ response: We agreed with the reviewer’s suggestion. Thus, we rewrote the sentences in the Results section. In the Results section, we omitted the sentences (lines 223- 226, 244-248, 262-263, 277-281).

Round 2

Reviewer 1 Report

Comments and Suggestions for Authors

The authors have put efforts to improve their manuscript in terms of clarity and presentation. I recommend the publication of the corrected manuscript in Foods.

Author Response

Thank you for your comments. Your suggestions significantly improved our manuscript.

Reviewer 2 Report

Comments and Suggestions for Authors

Authors should perform more experiments to prove their hypothesis.  

The authors have not addressed all the concerns. 

Comments on the Quality of English Language

fine 

Author Response

Thank you for your comments and suggestions. Point-to-point responses are as follows.

  1. Authors should perform more experiments to prove their hypothesis.

We agree that some of the experiments presented by the reviewer need to be examined in the future. Still, the previous revision period was 10 days, and this one was also 5 days, which is a very short time, and it is impossible to conduct the experiments again. We also think it is out of line with the editorial policy of this journal.

  1. The authors have not addressed all the concerns.

In the previous revised manuscript, we answered all the questions from the reviewer. While we could not determine MBC due to solubility issues with the ester compounds studied, we performed a Time-kill assay on the most soluble Pru-C12. We found that 2 hours of Pru-C12 treatment resulted in the disappearance of viable bacteria. The new results show that Pru-C12 has no viable bacteria after 2 hours of treatment, indicating bactericidal activity.

Second, in biofilm formation inhibition experiments, the method used in this paper is widely practiced worldwide and has been employed in many previous papers. The authors do not have a fluorescent confocal microscope at their research facility, making the experiment impossible.

As for the cytotoxicity test, I see no rational reason why HEK293 cells should be used. As noted in the previous reply, cytotoxicity studies using HeLa cells have been conducted for many years and do not appear problematic. This experiment was performed to select candidates with low cytotoxicity for animal studies from more soluble compounds.

Also, the reason for writing the results and discussion separately is that, as mentioned in the previous revision, we wanted to discuss each result carefully.
